# Surfactants, Biosurfactants, and Non-Catalytic Proteins as Key Molecules to Enhance Enzymatic Hydrolysis of Lignocellulosic Biomass

**DOI:** 10.3390/molecules27238180

**Published:** 2022-11-24

**Authors:** Salvador Sánchez-Muñoz, Thércia R. Balbino, Fernanda de Oliveira, Thiago M. Rocha, Fernanda G. Barbosa, Martha I. Vélez-Mercado, Paulo R. F. Marcelino, Felipe A. F. Antunes, Elisangela J. C. Moraes, Julio C. dos Santos, Silvio S. da Silva

**Affiliations:** 1Bioprocesses and Sustainable Products Laboratory, Department of Biotechnology, Engineering School of Lorena, University of São Paulo (EEL-USP), Lorena 12.602.810., Brazil; 2Biopolymers, Bioreactors, and Process Simulation Laboratory, Department of Biotechnology, Engineering School of Lorena, University of São Paulo (EEL-USP), Lorena 12.602.810., Brazil

**Keywords:** lignocellulosic biomass, surfactants, biosurfactants, non-catalytic proteins, enzymatic hydrolysis

## Abstract

Lignocellulosic biomass (LCB) has remained a latent alternative resource to be the main substitute for oil and its derivatives in a biorefinery concept. However, its complex structure and the underdeveloped technologies for its large-scale processing keep it in a state of constant study trying to establish a consolidated process. In intensive processes, enzymes have been shown to be important molecules for the fractionation and conversion of LCB into biofuels and high-value-added molecules. However, operational challenges must be overcome before enzyme technology can be the main resource for obtaining second-generation sugars. The use of additives is shown to be a suitable strategy to improve the saccharification process. This review describes the mechanisms, roles, and effects of using additives, such as surfactants, biosurfactants, and non-catalytic proteins, separately and integrated into the enzymatic hydrolysis process of lignocellulosic biomass. In doing so, it provides a technical background in which operational biomass processing hurdles such as solids and enzymatic loadings, pretreatment burdens, and the unproductive adsorption phenomenon can be addressed.

## 1. Introduction

In the current situation of energy and environmental crisis, many efforts have been made to find alternative sources to obtain fuels and commodities different from those from oil-based refineries [1]. Lignocellulosic biomass (LCB) is an attractive feedstock due to many factors: it is abundant and inexpensive, in addition to not competing with the food supply chain [2].

Lignocellulosic biomass is composed of lignin and two carbohydrate polymers from which it is possible to obtain fermentable sugars (e.g., glucose and xylose) to produce biofuels and high-value-added products in biorefineries. However, LCB has a complex and recalcitrant structure that resists the attack of microorganisms or enzymes [3]. To overcome this complex structural barrier, several physical, physicochemical, and biological strategies (pretreatments) have been used, playing an important role in the conversion of LCB. Nevertheless, there are still some bottlenecks in the enzymatic hydrolysis of LCB, such as non-productive binding, low total solid loading, and low enzyme activity. All those negative factors could be mitigated by the application of additives. In industry, additives play an important role in many areas, and in biorefineries, molecules such as surfactants have been effectively used to increase the saccharification yield of LCB [4]. In addition to surfactants, their biological counterpart (biosurfactants), and more recently non-catalytic proteins have shown a high potential to reduce the negative effects of the complex structure of the LCB against enzymatic access [5,6].

Thus, a process with effective pretreatments associated with subsequent enzymatic hydrolysis is the main path to integrate a second-generation technology in the current and well-established first-generation biorefineries. This review is focused on the role, mechanisms of action, and future strategies of using surfactants, biosurfactants, and non-catalytic proteins in the enzymatic saccharification step of LCB processing.

## 2. Structure, Lignin Recalcitrance, and Cellulose Crystallinity as Main Challenges to Valorize Lignocellulosic Biomass

Vegetal biomass is composed mainly of two carbohydrate polymers (cellulose and hemicellulose), a hydrophobic macromolecule (lignin), and other components (proteins, lipids, and inorganic compounds) [7]. Cellulose is the most abundant organic material in nature, being a linear polymer of high molecular weight, insoluble in water, comprising of cellobiose units linked jointly by β-1,4 glycosidic bonds. This polymer is constituted of highly amorphous regions, as well as joints and crystalline regions caused by the density of its compositional structure [8]. The number of glucose units in a cellulose molecule varies from 1000 to 50,000, depending on the plant’s origin. In cases of molecules with a high degree of polymerization, the cellulose molecules can aggregate together, due to hydrogen bonding and microfibrils, which are the building blocks of the fibers, acting on the cellulose fiber [9]. The degree of polymerization and the crystallinity are significant factors in the cellulose hydrolysis process. Another polysaccharide present in plant biomass is hemicellulose, acting as a reserve and support compound, thus being a complex polymer in terms of components and molecular structure. This compound is formed by a group of branched polysaccharides, consisting mainly of pentose, hexose, and uronic-acid residues, being classified according to the predominant carbohydrate in the main chain and the side branch [10]. The third main component of lignocellulosic materials is lignin, a complex amorphous polyphenolic macromolecule constituted mainly of phenylpropane units (p-coumaryl, coniferyl, and sinapyl alcohols) linked by carbon–carbon bonds. This component is associated with cellulose and hemicellulose, conferring rigidity and low reactivity to the set of plant fibers, and resistance to microbial degradation and enzymatic action [9,11]. These three components are linked intrinsically and the amount of each one varies according to biomass origin.

Among the many lignocellulosic materials, sugarcane bagasse, sorghum bagasse, rice husk, corn straw, rice straw, sugarcane straw, wheat straw, corn cob, barley straw, hardwood, and softwood stand out as potential and established sources to produce biofuels and value-added products in a biorefinery concept [12]. However, the recalcitrance of lignocellulosic materials is considered a challenge for the depolymerization of holocellulose (cellulose + hemicellulose) into its monomeric constituents such as glucose, xylose, and arabinose. Nevertheless, these sugars can be used as fermentable carbohydrates for conversion into biological products of commercial importance by microbial fermentation among high-value-added products or commodities [13]. An intensive process to obtain fermentable sugars would require an enzymatic step. This process could fractionate the lignocellulosic biomass and sequentially ferment the released sugars in biofuels or high-value-added products. Although lignin is an important structural component, this macromolecule is the principal barrier to valorizing lignocellulosic biomass via an enzymatic process. This component is one of the main challenges to be overcome in second-generation biorefineries due to different negative characteristics associated with the interaction between lignin and catalytic enzymes (e.g., cellulases). The principal well-described phenomenon associated with lignin during enzymatic hydrolysis is non-productive binding. The characteristics of different lignins associated with this phenomenon have been studied. Among these characteristics, the hydrophobicity and the content of guaiacyl, phenolics hydroxyl groups, and free phenolic groups present in lignin are identified as factors that favor the phenomenon of non-productive binding [14].

The above is the main reason for the use of pretreatments to process this recalcitrant material. Thus, pretreatment is the first step in obtaining fermentable sugars from lignocellulosic materials. There are different methods of pretreatment, such as chemical (acid, alkaline, organosolv, ionic liquids, supercritical fluids, etc.), physical (milling, microwave, pyrolysis, hot water, steam explosion, etc.), physicochemical (ammonia steam explosion, CO_2_ steam explosion, etc.), and biological (enzymatic) [15,16]. Among these approaches, alkaline pretreatment stands out because of its potential for lignin reduction, which makes feasible further processes of enzymatic saccharification. Many reagents have been used as catalysts in alkaline pretreatment technologies, such as calcium hydroxide, potassium hydroxide, aqueous ammonia solution, sodium hydroxide, and others, reducing the crystallinity of the biomass, separating structural bonds between lignin and carbohydrates, breaking the macromolecular structure, increasing the surface area of the biomass, and, hence, producing minimal inhibitory subproducts, and enhancing biomass enzymatic digestibility [11]. Nevertheless, pretreatments count for around 30–40% of the total cost of the process, so alternative strategies must be considered to make it viable [17]. In addition to pretreatments, an alternative approach is the use of additives (e.g., surfactants, biosurfactants, and non-catalytic proteins) to enhance the enzymatic process and valorize the lignocellulosic biomass in a biorefinery concept.

## 3. Additives as Emerging Tools to Enhance Enzymatic Hydrolysis

### 3.1. Surfactants

Surfactants, amphiphilic molecules capable of reducing surface tension and modifying the structure of plant biomass, can be used as additives in the enzymatic hydrolysis of vegetal biomass to improve the process efficiency. Studies have reported an increase in the efficiency of the enzymatic hydrolysis of various substrates by the addition of surfactants with variable properties, and some examples are described in Table 1. Likewise, different mechanisms that result in the beneficial effect due to the addition of surfactants on the enzymatic hydrolysis of cellulose have also been proposed.

One of the effects caused by the addition of surfactants is related to changes in the substrate structure, such as a reduction in the availability of lignin or an increase in the available cellulose surface [18]. Surfactants can be added in enzymatic hydrolysis, mainly non-ionic surfactants, to act as competitors for interaction with lignin and prevent the unproductive adsorption of cellulases or xylanases to lignin. Chen et al. [19] demonstrated that the addition of Tween 20 improved the efficiency of the enzymatic hydrolysis of wheat straw pretreated with acid, making it possible to obtain an 80% glucose conversion yield. According to the authors, the improvement in enzymatic hydrolysis was not only due to the blocking of lignin-cellulase interactions caused by the presence of the surfactant but also due to changes in lignin properties (e.g., hydrophobicity, hydrogen binding capacity, and surface charges) in the presence of the surfactant. Zhang et al. [20] also used surfactants to improve the enzymatic hydrolysis of metal-salt-catalyzed pretreated sugarcane bagasse. By adding Tween 80 during enzymolysis, all substrates pretreated with metallic salts showed higher glucose yields, reaching a maximum glucose yield (88.0%) with biomass pretreated with AlCl_3_. This result was obtained due to the reduction of unproductive adsorption caused by the interaction between Tween 80 and lignin, providing more cellulase for the process.

The action of surfactants in reducing the unproductive adsorption of cellulases may also be due to their influence on the desorption behavior of enzymes to lignin [21]. The influence of Tween 80 on the desorption of cellulases (Celluclast 1.5 L and Novozyme 188) bound to the acid-insoluble lignin present in corn husks was demonstrated by the reduction in yield from 43.6% to 21.5% of biomass hydrolysis performed by the lignin-enzyme complex in the presence of the surfactant [22]. This decrease in hydrolysis yield was due to the lower availability of enzymes for hydrolysis due to the desorption of enzymes from lignin, which may be related to competitive adsorption between enzymes and Tween 80 on lignin. In addition to the effect of competition between surfactants and enzymes for interaction with lignin, the presence of surfactants can reduce the affinity between cellulases and lignin, which also contributes to increasing the desorption of these enzymes [23]. Therefore, considering the action of surfactants in depressing unproductive adsorption and promoting the desorption of enzymes linked to lignin, the addition of these compounds before or after the addition of enzymes can be beneficial for the hydrolysis of lignocellulosic biomass [22,24].

Another mechanism proposed to explain the beneficial action of surfactants in the enzymatic saccharification process is related to the increase in cellulase enzyme stability and reduction in protein deactivation by thermal factors or shear forces [25,26]. The addition of Tween 20 has been shown to increase by about 2 °C the enzyme deactivation temperature by improving the thermal stability of cellulases from *Trichoderma reesei* [23]. Okino et al. [27] also observed that an increase in the rate of enzymatic hydrolysis by the addition of Tween 80 was possible only under stirring, suggesting that the use of this surfactant helps in the stabilization of unstable cellulase components under stirring.

Despite the positive effect of the addition of surfactants on enzymatic hydrolysis, care must be taken with the type of surfactant and the concentration to be used in the process. Ahammad et al. [28] observed that increasing the concentration of different surfactants from 5 to 8% did not result in an improvement in glucose yield but did result in a decrease in sugar release and possible inhibition of enzymatic activity. These results highlight the importance of developing studies to optimize the type and concentration of the surfactant that should be used in the process to obtain high yields of glucan conversion. In addition to surfactants, other compounds with surfactant action, such as some polymers, can also be used to improve the efficiency of the enzymatic hydrolysis of lignocellulosic materials.

#### Molecules with Surfactant Properties: Polymers

Several studies propose the use of molecules with surfactant properties to improve the enzymatic hydrolysis of lignocellulosic materials [29]. For example, a study aimed at developing an optimized model for low-cost enzymatic hydrolysis of hydrothermal-alkaline/oxygen pre-treated reed showed that it was possible to reduce the enzymatic load from 20 to 3 FPU/g of pre-treated biomass by adding 0.03 g/g-PS of polyethylene glycol (PEG) 3000 to the hydrolysis carried out with 2% (*w*/*v*) of solids content [30]. Under these optimized conditions, 94.5% of the xylan dissolution rate and 95.6% of glucan digestibility were obtained, with an increase of 30.7% in the glucan conversion rate and a saving of 85% in cellulase load compared with that in the treatment without additives.

In addition to reducing the enzyme loading, other effects caused by the presence of different polymers with surfactant properties are also suggested in the literature. For example, during enzymatic hydrolysis, cellulases can form complexes with phenolic compounds present in the biomass, reducing the availability of the enzyme for biomass hydrolysis. In order to study the mitigation of phenolic-compound inhibition of enzymatic hydrolysis, Avicel (23 g/L) was hydrolyzed with cellulases (0.25 g/L), tannic acid (0.3 mM), and PEG 4000 (6.25–50 g/L). Using 50 g/L of PEG 4000, the efficiency of enzymatic hydrolysis was improved, as this polymer breaks down the enzyme–tannic acid complex and increases enzymatic activity, enabling 100% recovery of losses in the extension of hydrolysis after four days [31]. Another example is the effect of considerably reducing the precipitation of commercial cellulase preparations from *Trichoderma reesei* by the addition of PEG6000, which was suggested by Chylenski et al. [32] as a potential strategy to promote a better dispersion of cellulases during enzymatic hydrolysis. Furthermore, Lou et al. [33] observed that cellulases could be easily deactivated at high agitation rates, especially when the enzyme concentration is low. Thus, by adding PEG4600, which can protect enzymes against shear-induced deactivation, it was observed that, although nonionic surfactants could not improve the efficiency of enzymatic hydrolysis of Avicel at 0 and 100 rpm, it was possible to significantly increase enzymatic hydrolysis at a high agitation rate (200 and 250 rpm).

**Table 1 molecules-27-08180-t001:** Effect of several surfactants on the glucan conversion yield of enzymatic hydrolysis of lignocellulosic materials.

Biomass	Enzyme Loading	Solids Loading (%, *w*/*v*)	Time (h)	Conversion Yield (Glucose) %		References
				Without Additive	With Additive	
Dilute acid-pretreated wheat straw	15 FPU/g solid (Cellic CTec2)	5	100	~65.0	80.0 Tween 20 (5.0 g/L)	[19]
Steam-pretreated poplar pulp	5.625 FPU/g (Cellic CTec2)	5	96	40.0	57.0 TritonX-100 (5.0%)	[28]
				40.0	41.0 Tween 80 (5.0%)	
Ground acid-pretreated sugarcane bagasse	20 FPU/g cellulose (cellulase Imperial Jade Bio-Technology)	10	72	27.0	46.0 Tween 20 (0.5%)	[34]
AlCl_3_-pretreated sugarcane bagasse	20 FPU/g dry pretreated solids (Cellic CTec2)	2	72	70.0	88.0 Tween 80 (150 mg/g biomass)	[20]
Dilute acid-pretreated bamboo residue	20 FPU/g glucan (Cellic CTec2)	5	72	29.4	61.6 Tween 80 (0.3 g/g lignin)	[35]
Alkaline-pretreated sugarcane bagasse	40 FPU/g cellulose (cellulase) and 30 U/g hemicellulose (xylanase)	5	72	40.7	57.3 Tween 60 (2.0%)	[36]
				40.7	60.0 Tween 61 (0.5%)	
Acid-pretreated wheat straw	20 FPU/g (cellulase from *Trichoderma reesei* ATCC26924)	10	72	67.4	72.6 Tween 80 (53.5% on dry pulp)	[37]
				67.4	74.4 PEG 6000 (50% on dry pulp)	

Therefore, the challenges for carrying out enzymatic hydrolysis of lignocellulosic biomass are known and demand research to overcome them. The use of additives (surfactants and polymers) has been studied to improve the enzymatic process, with different suggestions of mechanisms of action of these molecules. However, studies are still needed to optimize the type of molecule and determine the ideal concentrations of additives, in addition to investigating new mechanisms of action of surfactants and possibilities for the use of other additives, such as biosurfactants.

### 3.2. Biosurfactants

Biosurfactants are molecules with structural characteristics similar to synthetic surfactants. They are amphiphilic molecules composed of hydrophobic and hydrophilic portions. In common with their synthetic analogs, biosurfactants have the property of reducing surface and interfacial tension [38,39]. Due to their properties, the surface-active molecules can be applied as additives in the enzymatic hydrolysis processes of complex materials [40,41]. However, synthetic surfactants are the most used, even though these substances are not biodegradable and considered toxic to the environment [42]. Thus, the use of biological surfactants of microbial origin represents a more sustainable condition since these are biodegradable, biocompatible molecules with low toxicity [41,43].

Studies show the use of these biomolecules in the enzymatic hydrolysis process and suggest that they act by reducing the unproductive adsorption of enzymes to lignin and hemicellulose, increasing enzyme activity and stability, to increase conversion efficiency (Figure 1) [40,41,44]. The main group of biosurfactants that have stood out in the bioconversion of lignocellulosic materials is rhamnolipids [45]. These are compounds that have hydrophilic rhamnose fractions linked to hydrophobic chains of β-hydroxylated fatty acids [46]. They have high biodegradability, and excellent surface activity, so it is suggested that they are used to prevent the non-productive adsorption of enzymes to lignin during the hydrolysis of lignocellulosic raw materials [42,43].

Mesquita et al. [5] evaluated the process of glucose release by enzymatic hydrolysis of eucalyptus chips used for ethanol production. During saccharification, the effect of rhamnolipid with 10% (*w*/*w*) solids and 14 FPU/g of the enzymatic load was studied. Under these conditions, it was observed that in the hydrolysis with biosurfactant the glucose concentration was 62% higher than that obtained without the addition of biosurfactant. Zhang et al. [42] suggested the action of rhamnolipid in increasing the activity of β-glucosidase and stability of cellulase (Cel7A), in addition, was to decrease the unproductive adsorption of enzymes during the saccharification of rice straw. They also suggest that the increase in enzyme activity after the addition of the biosurfactant favors the decomposition of cellulose into glucose. So biosurfactants not only increase the conversion of glucose but also reduce the amount of enzyme needed and promote its recycling process. Wang et al. [44] analyzed the stability and activity of Cellulase R-10 and its action in the saccharification process of wheat straw. The results suggest that rhamnolipid promoted a better effect on cellulase activity than Triton X-100. It was also observed that the effect on the activity stability and glucose conversion occurred until the determined concentration (60 mg/l of rhamnolipid). This condition can be explained by the way biosurfactants behave in solutions. In an aqueous solution, the surfactant molecules can be monomers, thus having the ability to adsorb on surfaces and interfaces. As the concentration of these molecules in the solution increases, there is a critical value of concentration at which is observed the formation of aggregates known as micelles. This point is called the critical micelle concentration, and above this threshold, when biosurfactants are added, they will interact with each other, forming micelles [47,48]. These micelles can act as carriers of substances or aid in the solubility of insoluble compounds.

In addition to rhamnolipids, other biosurfactants can be applied in enzymatic hydrolysis processes. Menon et al. [49] observed an increase in the hydrolysis conversion of the hemicellulose of wheat bran of about 20% when adding 1% (*m*/*v*) of sophorolipids. However, the results in xylan hydrolysis were lower, which may be due to the non-crystalline nature of the substrate and fewer non-active binding sites. Xu et al. [40,50] observed that the presence of sophorolipids increased the yield of sugarcane bagasse saccharification by 7.4% and 17.8%, respectively. As mentioned for other biosurfactants, it is suggested that sophorolipids can alleviate unwanted unproductive adsorptions between substrates and enzymes caused by hydrophobic and electrostatic forces. Hosny et al. [51] evaluated the ability to hydrolyze urban waste by cellulase-producing bacteria, using a bacterial biosurfactant as an additive. Thus, after selecting the biosurfactant-producing bacteria, the biomolecule was isolated and characterized as a lipopeptide (surfactin). When adding the mixture to the hydrolysis, the release of approximately 9076.1 ug/mL of sugar was observed, being 92.2% higher than the results obtained with Tween 80. In addition, greater activity of cellulase was observed in the presence of this biosurfactant. Therefore, it is observed that the use of these biomolecules as additives in the saccharification steps is promising, and they have positive effects on the final yields of fermentable sugars release. Thus, their use as additives in the saccharification of lignocellulosic biomass in biorefineries is of high environmental, technological, and economical interest.

### 3.3. Non-Catalytic Proteins

Other emerging boosters that can be used to enhance the saccharification process of lignocellulosic biomass are the non-catalytic proteins. Those proteins have been classified, according to their derived sources, as animal, plant, bio-non-hydrolytic protein, and others [52]. Non-catalytic proteins are currently proposed as lignin-blocker additives to avoid the non-productive adsorption phenomenon [6,53]. Nevertheless, other effects have been described, such as increasing cellulose activity, a reduction in particle size and crystallinity (during pre-hydrolysis steps), increasing enzyme accessibility (amorphogenesis), and decreasing cellulose interfacial tension [6,54,55,56]. The literature also highlights a group of non-catalytic proteins that include expansins, loosenins, and swollenins that act by breaking hydrogen bonds and altering the cellulose fiber structure [57,58].

Among the different proteins that have been used to enhance the enzymatic hydrolysis step, Bovine Serum Albumin (BSA) has been highlighted and used as a model due to its positive effects (e.g., alleviating the feedback inhibition of cellobiose, protecting the enzymes from thermal deactivation, and relieving the cumulative sugar inhibition, in addition to those already described) when used as an additive in the saccharification of different substrates [59]. However, BSA preparation and purification could increase the cost of the process. In this context, other alternative proteins have been studied to enhance the saccharification process of different lignocellulosic biomass, as can be seen in Table 2.

As described by many authors, the use of non-catalytic proteins during enzymatic hydrolysis has many operational and cost-effective advantages, as are observed with surfactants and biosurfactants. In this context, the integration of their mechanisms and properties could impact positively their use as formulations in a biorefinery concept.

### 3.4. Surfactants, Biosurfactants, and Non-Catalytic Proteins Integration to Enhance the Saccharification Process

Among the hurdles encountered in the enzymatic hydrolysis of biomasses being tackled by scientists and industry, there must be highlighted a few that pose a more significant constraint to the exploitation of high-efficiency biomass conversion; these are: (i) low solids loading during enzymatic hydrolysis [63]; (ii) inhibition of cellulosic enzymes by the generated products [64]; and (iii) the unproductive binding of enzymes to the lignin matrix [65,66].

With respect to the constrained values of total solids loading, some authors have gathered evidence that indicates that higher solids loading implicates, during enzymatic hydrolysis, an increase in the system’s viscosity, hampering heat and mass transfer between the biomass, enzymes, and the free water in the system; hence, decreasing the capacity for diffusing the enzymes, substrates, and the required conditions to adsorb/desorb at the surface of the lignocellulosic material [67].

In addition, the hydrolysis end-products such as the monomeric and oligomeric sugars (e.g., cellobiose, xylans, xylooligosaccharides, glucose, and others) and some inhibitors generated during the pretreatment steps (e.g., phenols, furfural, hydroxymethylfurfural, gluconic acid, and others), in a context of low water activity, could strongly bind to the catalytic sites of cellulosic and hemicellulosic enzymes [68]. Ultimately, the unproductive adsorption of the enzymes at the surface of lignin and pseudo-lignin complexes poses an overwhelming barrier due to the strong attachment at the catalytic and binding sites of the enzymes [69].

Nevertheless, as described in the previous sections, surfactants, biosurfactants, and non-catalytic enzymes are considered additives in enzymatic hydrolysis processes; they have certain physicochemical and biological properties which aid in the hydrolysis-associated issues. Therefore, the integration of these strategies to develop a formulation of additives to enhance hydrolysis must be cautiously conducted in order to obtain a clear response regarding the underlying mechanisms generated from the resulting complex interplay. In this context, it is important to underpin the reasons that make the aforementioned molecules promising alternatives to formulate potential hydrolysis cocktails and overcome the current technological constraints.

On the one hand, surfactants have been used as additives in biomass pretreatments for multiple purposes, and it is probable that tensoactive molecules bear the capacity to increase the contact between the reactants and the biomass, reduce the surface tension among the fluids and also ameliorate the detachment of lignin from the biomass [4]. Despite that, regarding their use in saccharification processes, depending on the type of surfactant, it is hypothesized that the capacity for altering the biomass surface hydrophobicity by loosening the crystalline structure of cellulose and opening pores through lignin may provide optimal conditions to facilitate the adsorption/desorption of substrate and end-products from the enzymes [19,70,71,72]. Moreover, surfactants could also increase the availability of free water by reducing the surface tension of air/solid/water and reducing the shear stress in the case of high solids loading [33,36,73].

In addition, biosurfactants can exhibit versatile properties that improve hydrolysis efficiency. One of the main advantages conferred by these molecules is the stabilization of proteins by establishing intermolecular interactions and conserving the structure and conformation of the enzymes due to the high potential zeta value of biosurfactants [74,75]. Moreover, they can also form micelles, which could serve as vesicles, diminishing the undesired binding of the catalytic sites to non-targets compounds and reducing the denaturation caused by shear forces [40]. Furthermore, these properties can cause an improvement in the enzyme activity that could indirectly enhance the saccharification yield as well as by interacting with cellular membranes, biosurfactants could augment the expression of genes involved in the synthesis of enzymes [75,76]. For example, it was observed by many researchers that microbial cells in the presence of rhamnolipids (class of glycolipids) exhibited changes in the cell surface functional groups [77,78,79]. These changes alter the structure and hydrophobicity of the cells, and this effect may also be explored in the case of saccharification processes, implicating amendments to the substrate functional groups as well as to the enzymes.

The path toward optimized hydrolysis formulations also requires the use of non-catalytic enzymes. As an example of a non-catalytic protein widely applied to saccharification processes, Bovine Serum Albumin (BSA) stands out [80,81]. Nonetheless, other soy-derived, peanut-derived, and whey proteins have been applied to enzymatic hydrolysis of different biomasses and positive effects were observed regarding lesser enzyme loading and higher hydrolysis yield in comparison to controls without proteins supplementation [6,41,82]. Non-catalytic proteins have the capacity to bind onto lignin due to terminal hydrophobic amino acids and chemical groups that interact with the aromatic rings restrained in the lignin skeleton and organic groups at the backbone, hence creating a steric and electrostatic barrier that reduces the adsorption and enables the other targeted molecules, such as cellulose and hemicellulose, to be catalyzed by the enzymes [41,83]. Likewise, by studying the isolated effect of sophorolipids (a class of glycolipids) on soy proteins, it was reported that the influence of the biosurfactant increased the α-helical structure of the protein, which favors its secondary conformation and possibly leads to a more flexible state due to the intermolecular interactions [76].

By integrating the aforementioned technologies and fundamentals, it is possible to tailor processes and strategies to mitigate the major drawbacks encountered in this field. As described, in a complex system containing biosurfactants, surfactants, non-catalytic proteins, cellulolytic enzymes, and biomass, one must evaluate the molecular interactions among each compound, and, most importantly, gather enough data to forecast the system as a whole. Within this context, Figure 2 exhibits a schematic representation of the hypothesized mechanisms occurring simultaneously during a developed enzymatic hydrolysis process conducted with formulations containing the presented compounds.

To gain more insights, only a few studies are found regarding the use of complex hydrolysis formulations [4,84]. More precisely, most works are conducted either solely with surfactant mixtures or non-catalytic proteins [6,34,84,85]. As yet, there have been few studies conducted with biosurfactants and even with the combination of surfactant/biosurfactant/non-catalytic proteins [41,51]. For example, Xu et al. [51] have combined different surfactants (e.g., ionic and non-ionic) with sophorolipid and whey protein to avoid unproductive adsorption, allowing the utilization of 20% (*w*/*v*) of total solids and an enzyme dosage of 4 FPU/g for hydrolysis. Markedly, in these conditions, a hydrolysis yield of 80% and a final glucose titer of 122 g/L were achieved.

The utilization of these active compounds was corroborated by the necessity of a system with a uniform dispersion of solids promoted by the ionic surfactant; with minimal shear forces acting upon the enzymes, which was conferred by the non-ionic surfactants; maximal stability and activity of the enzymes performed by the addition of the sophorolipid (biosurfactant); and, ultimately, the reduction in the unproductive adsorption of enzymes onto lignin, which was avoided by the addition of non-catalytic proteins.

Therefore, technologies regarding complex formulation development for the enhancement of enzymatic hydrolysis aimed at a modern biorefinery platform of multiple products are still showing promising results. However, surveys focused on the dosages, optimization, and integrated effect of each compound are still under-exploited. Thus, it is envisaged that advances in this field will not only contribute to the provision of robustness and cost-effectiveness of the process but also address sustainable solutions to environmental issues associated with the use of molecules derived from fossil and non-renewable sources, which is in agreement to the agenda of 2030 and its respective goals established by the United Nations Organization (UNO).

## 4. Potential Enzyme Cost Reduction and Future Perspectives in the Use of Additives in the Enzymatic Hydrolysis Process

With the advent of the biorefinery concept, lately, there has been an increasing interest in the enzymatic hydrolysis of lignocellulosic materials using microbial cellulases and xylanases to obtain second-generation ethanol and other value-added bioproducts [86]. Nevertheless, the need for a pretreatment method is a critical factor associated with the techno-economic analysis of the whole process [87]. As mentioned above, several molecules can be used as additives in enzymatic hydrolysis processes such as surfactants, biosurfactants, and non-catalytic proteins. Studies have demonstrated an improvement in enzymatic hydrolysis with the addition of these additives [20,41,88,89], and some authors mentioned the possibility of reducing enzyme dosage, which would improve the economic efficiency of the process [6,44,90]. The evaluation of additives combined with different amounts of enzymes has been reported to elucidate the effects. Zheng et al. [59] evaluated the impact of Tween 20, Tween 80, and BSA on enzyme loading and achieved the most effective results with Tween 20, with a two-fold reduction in enzyme loading. Similarly, Sanchez-Muñoz et al. [84] achieve a four-fold reduction using Tween 20 and PEG400 in the hydrolysis of sugarcane bagasse. In addition, more than a six-fold reduction in enzyme loading was achieved by Lu et al. [31] with a higher glucan conversion after 36 h of hydrolysis using PEG3000. The impact of non-catalytic proteins has been also evaluated by Brondi et al. [91], who tested different non-catalytic proteins, finding soybean protein to be the most effective with a reduction of 50% and 66% in enzyme loading and operational time, respectively. Moreover, a two-fold increase in glucose release was attained with BSA. These studies demonstrate the promising impact of additives on enzyme loading, which could significantly reduce enzyme costs.

However, the application of these additives needs optimization to obtain a positive impact on the process cost. As biosurfactants have a higher production cost than chemical surfactants and some non-catalytic proteins, the utilization of biosurfactants is more expensive, which the enzyme reduction does not compensate for in terms of cost. Mesquita et al. [5] evaluated the cost reduction in the overall process of ethanol production using surfactants in each step (i.e., feedstock, pretreatment, hydrolysis, saccharification, and fermentation) and found that the addition of biosurfactants in the hydrolysis step negatively affected the overall costs (e.g., rhamnolipids). Conversely, Tu and Saddler [92] achieved a reduction of 60%, 24%, and 8.6% in the enzyme cost, material cost, and process cost, respectively, with the addition of Tween 80 in the hydrolysis of pretreated softwood, demonstrating the cost difference in the utilization of additives. To overcome this low economic performance, more studies are needed focusing on the application of biosurfactants to enhance the operational conditions for the implementation of these additives with economic viability. Moreover, it is of fundamental importance to deepen the studies of developed strategies to make the overall enzymatic processes more cost-attractive for future industrial applications and the realization of a biorefinery concept.

Bearing in mind the advantages of using additives in enzymatic reactions, more studies are needed for a better understanding of the molecular dynamics of this process and the influence of physicochemical factors, such as pH, temperature, and ionic strength, on enzymatic mechanisms. Enzyme pool studies can also be performed in the presence of one or more additives to evaluate their effect on the hydrolytic activity. Furthermore, optimization of the experiments is necessary to perform a screening of variables and establish levels for these selected variables aimed to obtain enzymatic hydrolysis with the incorporation of additives in a cost-effective process.

## 5. Conclusions

Second-generation biorefineries represent the future of energy and commodities for many developed and undeveloped countries. To reach this goal, many strategies have been developed to make feasible the processing of lignocellulosic biomass (LCB). In the biorefinery concept, additives have been highlighted because of their potential to improve the enzymatic saccharification of LCB. Surfactants have been reported as one of the most effective additives due to their many advantages when applied in bioprocesses, such as decreasing the phenomenon of non-productive binding, enhancing enzyme stability and activity, and improving the saccharification yield. In addition, biosurfactants and non-catalytic proteins are emerging molecules with properties similar to surfactants. However, their production and purification are still a current challenge for large-scale technologies. Therefore, advances in the deployment of additives formulation, as well as their interplay with the enzymes, should be further conducted to identify reliable outcomes.

## Figures and Tables

**Figure 1 molecules-27-08180-f001:**
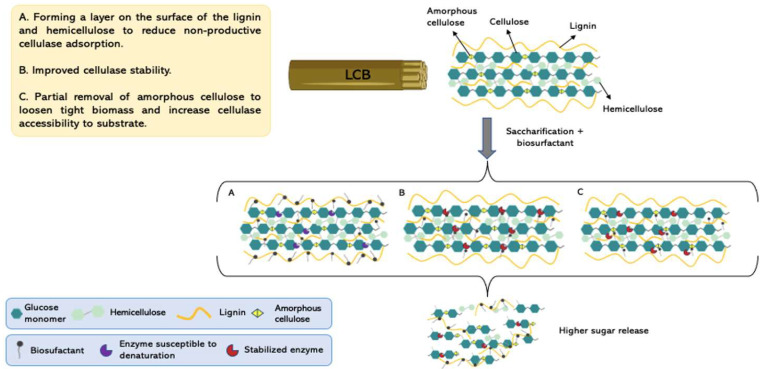
Underlying mechanisms of action of biosurfactants on pretreated lignocellulosic biomass and their effects during enzymatic hydrolysis (Created with BioRender.com).

**Figure 2 molecules-27-08180-f002:**
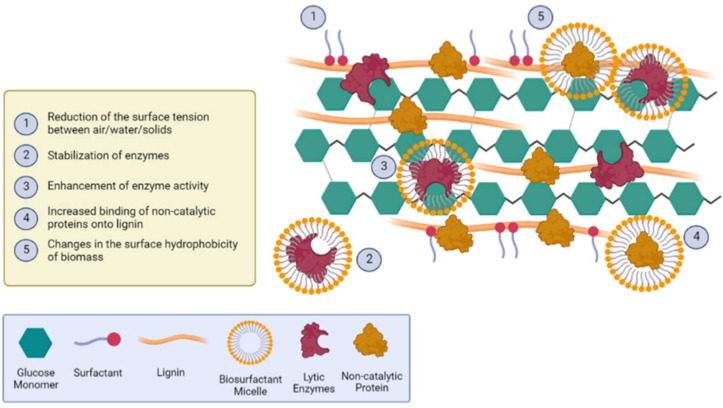
Schematic representation of the likely mechanisms occurring during the enzymatic hydrolysis of lignocellulosic biomasses supplemented with additive formulations composed of non-catalytic proteins, surfactants, and biosurfactants (Created with BioRender.com).

**Table 2 molecules-27-08180-t002:** Non-catalytic proteins as additives to enhance the enzymatic hydrolysis of lignocellulosic biomass.

Non-Catalytic Protein	Enzymatic Hydrolysis Conditions	Substrate (Pretreatment)	Increase in Conversion Yield	Reference
BSA 0.1 g/g dry solid	** Non-catalytic protein pretreatment* All experiments were carried out for 1 h at 50 °C using 50 mM citrate buffer (pH 4.8); Enzymatic hydrolysis was carried out for 72 h at 50 °C. Enzyme formulation was composed of β-glucosidase (15 CBU/g glucan) and cellulase (15 FPU/g glucan) and a TS loading of 8% (*w*/*w*).	Pretreated creeping wild ryegrass (Acid-H_2_SO_4_)	10.6%	[59]
BSA 50 mg/g of dry solid	Enzymatic hydrolysis was carried out for 48 h at 50 °C. Enzyme formulation was composed of cellulase (10 FPU/g of DM) and β-glucosidase (500 nkat/g DM). Xylanase was dosed at 0.18 mg/g DM. The TS loading was 1% (*w*/*v*).	Pretreated corn stover (ammonia)	31.37%	[60]
BSA, CSL, YE, and P 1.0 mg/mL (total protein)	** Non-catalytic protein pretreatment* All experiments were carried out for 12 h at 50 °C using 50 mM citrate buffer (pH 4.8); Enzymatic hydrolysis was carried out for 72 h at 50 °C using an enzyme dosage of 15 FPU/g of biomass and a TS loading of 2% (*w*/*v*).	Pretreated rice straw (alkaline-NaOH)	BSA-19.7% CSL-12.7% YE-13.5% P-13.7%	[61]
Soybean protein 4% (*w*/*w*)	1:1 cocktail from *A. niger* and *T. reesei.* All experiments were carried out for 24 h at 50 °C using 50 mM citrate buffer (pH 4.8) and a TS loading of 5% (*w*/*v*).	Pretreated sugarcane bagasse (Steam-explosion)	54%	[53]
Peanut protein 2.5 g/L (total protein)	** Non-catalytic protein pretreatment* All experiments were carried out for 2 h at 50 °C using 50 mM acetate buffer (pH 5); Enzymatic hydrolysis was carried out for 72 h at 50 °C. Enzyme formulation was composed of β-glucosidase (10 CBU/g glucan) and cellulase (5 FPU/g glucan) and a TS loading of 2% (*w*/*v*).	Pretreated bamboo (phenylsulfonic acid)	147%	[6]
Amaranth proteins 8% (*w*/*w*)	All experiments were carried out using 0.2 M Na-acetate buffer (pH 4.8) for 48 h at 50 °C. Enzyme formulation was composed of cellulases at 10.60 FPU/g biomass and xylanase at 6.72 U/g biomass, and a TS loading of 5% (*w*/*v*).	Pretreated Amaranth straw (liquid hot water)	12%	[62]

* *Non-catalytic protein pretreatment* is a pre-hydrolysis process to enhance the interaction between the non-catalytic proteins and the lignocellulosic biomass before the addition of the catalytic enzymes; CSL: Corn Steep Liquor; YE: Yeast Extract; P: Peptone; TS: Total solids; DM: Dry matter.

## Data Availability

The data presented in this study are available in this manuscript.

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
