# Peer review of "Surfactants, Biosurfactants, and Non-Catalytic Proteins as Key Molecules to Enhance Enzymatic Hydrolysis of Lignocellulosic Biomass"

_molecules, 2022, doi:10.3390/molecules27238180_

Round 1

Reviewer 1 Report

My only  concern is with regards to the language. The manuscript could use some editing or proofreading by a native speaker. Occasionally sentences are cut short or halved by parentheses or other punctuation, which makes the reader wonder what was it that the writer was trying to convey.

A fair proportion of the manuscript focuses on non-catalytic proteins for ehnancement of enzymatic hydrolysis of LCB. I feel that one (two) important class was missing from the list - expansins/expansin-like proteins. I understand that the literature is perhaps not too extensive on this topic, but at least a mention of this class of proteins would benefit the review and would give a more complete picture of the state of the art.

Over all, this is a really nice and well referenced review on a topic that is gaining more and more attention.

Reviewer 2 Report

The review "Surfactants, biosurfactants, and non-catalytic proteins as key molecules to enhance enzymatic hydrolysis of lignocellulosic biomass" is devoted to the description and effects of additives in the process of enzymatic hydrolysis of pretreated lignocellulose. Types of additives are listed: surfactants, biosurfactants, and non-catalytic proteins. The main idea of this review is undoubtedly suitable for Molecules. The review itself is quite rare and has no predecessors, which characterizes the authors very positively. The abstract clearly reflects the content of the review, but needs the following amendments: firstly, the statement "lack of technology for large-scale processing of lignocellulose" is false, secondly, "use of additives" is not a new strategy, but rather a good technique for successful enzymatic hydrolysis; thirdly, the use of such additives is possible only for pretreated lignocellulose. Therefore, the annotation needs to be corrected.

The title of the review includes the words “key molecules” and refers specifically to supplements. Enzymes are named key molecules in the annotation. This game of terms misinforms the reader. If the title is changed to "Surfactants, biosurfactants, and non-catalytic proteins enhance enzymatic hydrolysis of lignocellulosic biomass" the review will not suffer.

Notes:

1. Rewrite the annotation, eliminating the shortcomings.

2. Change the name so that the phrase "key molecules" remains only for enzymes, as it really is.

3. To cite, with the accompanying text, reviews on the enzymatic hydrolysis of lignocellulose with an emphasis on the fact that the key point in the technology of enzymatic hydrolysis of lignocellulose is pre-treatment, which makes it possible to destroy the composite structure of lignocellulose and partially remove non-cellulosic components (lignin, sometimes hemicelluloses).

For example,

Lee, S. Y., Kim, H. U., Chae, T. U., Cho, J. S., Kim, J. W., Shin, J. H., ... & Jang, Y. S. (2019). A comprehensive metabolic map for the production of bio-based chemicals. Nature Catalysis, 2(1), 18-33. https://doi.org/10.1038/s41929-018-0212-4

Chandel, H., Kumar, P., Chandel, A. K., & Verma, M. L. (2022). Biotechnological advances in biomass pretreatment for bio-renewable production through nanotechnological intervention. Biomass Conversion and Biorefinery, 1-23. https://doi.org/10.1007/s13399-022-02746-0

Pan, S., Zabed, H. M., Wei, Y., & Qi, X. (2022). Technoeconomic and environmental perspectives of biofuel production from sugarcane bagasse: Current status, challenges and future outlook. Industrial Crops and Products, 188, 115684. https://doi.org/10.1016/j.indcrop.2022.115684

4. It is unacceptable to call the components of lignocellulose "fractions". Need to fix.

5. There are no examples of using lignocellulose containing cellulose with a polymerization degree of 50,000 for enzymatic hydrolysis. As a rule, in pre-treated lignocellulose, the degree of polymerization of cellulose does not exceed 3000.

6. The description of Figure 1 should include a statement that the lignocellulosic biomass has already been pretreated. All of these additives will not affect the efficiency of enzymatic hydrolysis if native biomass is used as a substrate.

Reviewer 3 Report

Surfactants, biosurfactants, and non-catalytic proteins as key 2 molecules to enhance enzymatic hydrolysis of lignocellulosic 3 biomass

The current review describes the mechanisms, roles, and effects of using additives, such as surfactants, biosurfactants, and non-catalytic proteins, separately and integrated into the enzymatic hydrolysis process of lignocellulosic biomass.

The review is interesting.  My own comments are given follow:

1. Abstract need more improvements.

2. The importance of the review should be highlighted for readers

3. Some sections needs more explanations mainly :

- 3.1.1 Molecules with surfactant properties: polymers

- 3.3. Non-catalytic proteins

4. Format of table is not attractive

5. More references should be added

6. The authors should give future recommendations for this review in the conclusion part
